

# Reconstruction of winter temperature of southwest China over the past 300 years based on a Bayesian approach

Siyu Chen[1,2], Stefan Brönnimann[1,2]

[1]Institute of Geography, University of Bern, Bern, 3012, Switzerland
[2]Oeschger Centre for Climate Change Research, University of Bern, Bern, 3012, Switzerland

*Correspondence to*: Siyu Chen (siyu.chen@unibe.ch)

**Abstract.** A Bayesian approach was applied to reconstruct winter temperatures in southwestern China from 1700 to 1949, using 1950–1999 as the reference period. Within this methodological framework, documentary data provided weather and climate information to generate the Cold Weather Index (CWI; 1 = warm, 6 = extremely cold), two paleoclimate simulation ensembles served as priors, and uncertainties in the documentary records together with the dependence of observations on climate contributed to the likelihood estimates. The reconstructed CWI identified 20 extremely cold, 34 cold, and 15 warm winters, with the 1890s emerging as the coldest decade of the past three centuries due to a succession of severe winters. The posterior reveals pronounced interannual variability with an amplitude of ~2.77 °C, slightly smaller than existing reconstructions from an individual station, where the coldest winter was ~2 °C colder than the reference period. On longer timescales, the reconstruction captures a cold phase in the latter part of the nineteenth century and the warming in the twentieth century. An alternative reconstruction using time-independent priors demonstrate the capacity of the approach to disentangle the contributions of simulations and documentary evidence. This study provides a new regional climate reconstruction for southwestern China and highlights the potential of Bayesian approach for obtaining climate reconstructions from documentary climate data.

## 1 Introduction

Southwest China (97°E–110°E, 20°N–34°N) includes three provinces—Sichuan, Guizhou, and Yunnan—and one municipality, Chongqing. It is a data-sparse region characterized by complex climate and topography. Its distance from the Mongolia–Siberia cold source region and the rising elevation from east to west reduce this region's exposure to cold waves, with southwestern Yunnan being particularly unaffected (Y. Ding et al., 2013). As a result, both data limitations and natural complexity make investigations of past climate change more challenging. In the 1970s, Yu (1996) discussed long-term climate changes using historical records of snow cover on Cang Mountain in western Yunnan province. Yang (2007) developed a regression model to reconstruct annual winter mean temperature in Kunming (the capital city of Yunnan province) from 1721 onward based on snowfall and precipitation indices in the dry season. Bi et al. (2020) generated regional winter temperature indices (from -2 to 2) during the Qing dynasty (1644-1911) based on information from local gazetteers. Additionally, tree-ring





records also reveal variations of annual or winter temperatures primarily from the western part of Sichuan and Yunnan province which are close to the Tibetan Plateau (Fan et al., 2008; Keyimu et al., 2020; Shi et al., 2017; X. Zhang et al., 2016). Nevertheless, region-wide, high-resolution reconstructions of winter temperature in southwest China are still remains necessary and importance, compared to the data-rich eastern China (Z. Hao et al., 2018).

Documentary data is one of the main proxies for climate reconstruction and is characterized by accurate dating, high spatial
coverage, and high information content. Based on different types of documentation (e.g., diaries, archives, disaster chronologies) and reconstruction approach (e.g., regression function, index), it is possible to reconstruct past climate change from annual, decadal to multidecadal scale (Zheng et al., 2014). The "index" approach depicts the temperature or precipitation departure from climatology, by converting descriptive data into indices usually symmetric (Chinese Academy of Meteorological Sciences, 1981; Nash et al., 2021). Generally, indices closer to climatology have higher probabilities, while
more extreme deviations are less likely. For example, Pfister et al. (2018) used a 7-scale index with equal probabilities (16.66%) for indices from -2 to 2, but only 8.33% for -3 and 3. Paleoclimatologists can also translate the climate indices or other key variables (e.g., snowfall days) into temperatures using regression models—a widely applied method with strong reconstruction skill (Z.-X. Hao et al., 2012; Zheng et al., 2018). However, this approach has strict requirements for the quality and continuity of historical observations. Furthermore, such backward models (predicting a climate variable from its effect) differ from reality,
where the documentary evidence depends on climate and not vice versa. If a forward model can be formulated, the Bayesian approach can then be used for inversion. It can maximize the use of a wide range of weather and climate-related information, including both qualitative and quantitative, and incorporate the uncertainty of the sources even for absence of evidence. It allows climatologists and historians to use climate simulation data as a prior and estimate the likelihood of each index to obtain posterior probability distributions. It has been applied in climatology field research, such as reconstructing seasonal
climatology for the Burgundian Low Countries in the 15th century (Camenisch et al., 2022).

This study aims to conduct a new application of the Bayesian approach to 1) interpret the historical weather and climate-related information and present a new winter temperature reconstruction of southwest China and 2) discuss the feasibility, potential, and limitations of the advanced Bayesian approach.

## 2 Data

In this study, we used three types of data for reconstruction: documentary data, paleo-simulations, and instrumental data. The documentary data contain information about winter climate or meteorological events over the past 300 years, expressed through cold winter indices. Paleo-simulations provided a climate background where all the information into which all information was assimilated. The instrumental data is used in temperature-precipitation relationship analysis during the past decades and aided in determining likelihood. We also used several other reconstructions for comparison and evaluation.



## 2.1 Documentary data

This study utilizes two types of documentary data: local gazetteers and *Yu-Xue-Fen-Cun*.

The local gazetteers documented disastrous events or unusual phenomena, typically in chronological format. They are often used to reconstruct dry/wet and cold/warm indices because they emphasize deviations from normal conditions and rarely record ordinary weather. The main sources of local gazetteer records are three compilations: *Natural Hazards Information Database of the Qing Dynasty* (Fang et al., 2020; Xia, 2015), *A Compendium of Chinese Meteorological Records of the Last 3000 Years* (D. Zhang, 2013), and *The Encyclopedia of Meteorological Disasters in China: Guizhou, Sichuan, Yunnan and Chongqing Volume* (Liu, 2006; Luo, 2006; Ma, 2008; Zhan, 2006). The first two compilations cover records until 1911, while the last extends to 2000. Besides, we extracted records from several other compilations as supplementation and validation (Guizhou Local Gazetteer Compilation Committee, 2016; Guizhou Provincial Library, 1981; Qin & Yu, 2001; Sichuan Local Gazetteers Compilation Committee, 1996, 1999; Yunnan Institute of Water & Hydropower Engineering Investigation, 2008; Yunnan Local Gazetteers Compilation Committee, 1995). The series of *Annual Report on the Phenology of Animals and Plants in China* (Institute of Geography in the Chinese Academy of Sciences, 1965, 1977a, 1977b, 1982, 1986a, 1986b, 1988a, 1988b, 1989a, 1989b, 1992) were used as supplement providing information on flowering periods, hereafter referred to as the *Phenology Report*.

The *Yu-Xue-Fen-Cun* archive is a unique piece of literature in China that contains summaries of monthly precipitation, reported by local governments to the Emperor during the Qing Dynasty. The archive often includes the date/frequency and amount (moisture penetration depth or snow depth) of rainfall or snowfall. Compared to the local gazetteers, Yu-Xue-Fen-Cun are more quantitative and systematic and, therefore, suitable for reconstructions of absolute values of climate elements (Q.-S. Ge et al., 2005; Z.-X. Hao et al., 2012). In this study, we utilized the *Yu-Xue-Fen-Cun* archive from the Qing Dynasty Archives Retrieval System of the National Palace Museum, Taipei (https://qingarchives.npm.edu.tw/), three compilations (The First Historical Archives of China, 1984, 1991, 1996) and the records provided in appendix of Yang's book (2006). An example of *Yu-Xue-Fen-Cun* archive of December in 1783 (the 47th year of the Qianlong reign) from the Qing Dynasty Archives Retrieval System of the National Palace Museum, Taipei is shown in Fig. S1.

### 2.2 Paleo-simulations

We use two sets of climate model simulations. ModE-Sim is a medium-sized ensemble of atmospheric simulations comprising three sets of 20 members spanning 1420–1850 and two sets of 36 members spanning 1850–2009. We took 20 members from each period as the climatic prior. This ensemble is based on the ECHAM 6.3 model (T63; approx. 1.8° horizontal) with radiative forcings, volcanic forcings, and prescribed sea surface temperatures and sea ice (Hand et al., 2023). To assess the independence of our reconstruction from the prior, we also employed 13 all-forcing members of coupled simulations, with a horizontal resolution of 1.9°× 2.5°, spanning 850 to 2005, from the Community Earth System Model Last Millennium



Ensemble (CESM-LME) (Otto-Bliesner et al., 2016). Surface temperatures (TREFHT in CESM-LEM) between 1700-2000 from these two ensembles have been extracted, and the regional winter mean temperatures have been calculated. However, there is a large bias between these two ensembles which is local and latitude dependent (Fig. S2).

## 2.3 Instrumental data

The mean winter temperatures since 1901 were extracted from CRU TS 4.07v ([https://www.uea.ac.uk/groups-and-centres/climatic-research-unit/data](https://www.uea.ac.uk/groups-and-centres/climatic-research-unit/data)). For precipitation, we used Global Surface Summary of the Day (GSOD) dataset ([https://www.ncei.noaa.gov/data/global-summary-of-the-day/](https://www.ncei.noaa.gov/data/global-summary-of-the-day/)) containing 73 stations from 1942 to 2019 to derive daily precipitation type based on the field "FRSHTT", which indicates the occurrence of the following weather phenomena: fog, rain or drizzle, snow or ice pellets, hail, thunder and tornado/funnel Cloud. When using GSOD dataset, only winter with no
more than 5 days of missing value was involved in the correlation analysis. Additionally, we utilized a gridded dataset of daily precipitation types (rain, snow, sleet) at a $0.5° \times 0.5°$ resolution between 1961 and 2016 (referred to as DFDP), derived based on the different phase precipitation separation parameterization scheme (Su & Zhao, 2022).

## 2.4 Others

We used following reconstructions for comparison: ModE-RA is a set of paleo-reanalysis based on a Kalman filter approach
which assimilated information from both natural and documentary proxies and early instrumentals in to ModE-Sim. ModE-RAclim is a time-independent version of ModE-RA, which can be used to distinguish the effects of observations and simulations (Valler et al., 2024). The Eurasian winter temperature reconstruction named WinTEDA blended information from tree ring, ice core and early instrumentals into CESM-LME based on off-line assimilation (Tejedor et al., 2024).

About administrative division data, we maintain consistency with the current administrative divisions at the provincial level.
At the prefectural and county levels, to minimise the impact of administrative boundary changes on extracting observation locations, we use the CHGIS V6 Time Slice (1911) dataset for extracting observation locations, analysing and plotting (CHGIS, 2016).

## 3 Methods

### 3.1 Interpretation of the documentary data

### 3.1.1 Weather and climate-related information from local gazetteers

Weather- and climate-related information recorded in local gazetteers typically describe unusual phenomena or disastrous events and can be categorized into four subsystems: atmospheric, hydrological, biological, and human perception of cold/warm. Atmospheric anomalies include the precipitation (rain, snow, frost, freezing rain) and weather (prolonged cloudy or sunny). Hydrological anomalies refer to the frozen of water bodies and biological anomalies are the status of plants, animals, and



human beings (e.g. damaged, death). Human perception involves subjective descriptions of temperature such as "warm as spring" or "extreme cold". We extracted information on 10 types of unusual phenomena as follows: snowfall (SN), freezing rain (FR), frozen water (FF), frost (FT), precipitation (PR), plant damage (PD), animal death (AD), human death (HD), flower bloom (FB) and human perception (HP).

### 3.1.2 Precipitation information from *Yu-Xue-Fen-Cun*

Due to the Limited resolution of precipitation information in *Yu-Xue-Fen-Cun* for the three Sichuan, Yunnan, and Guizhou provinces (the Chongqing municipality was part of Sichuan province before 1997), the following information can be extracted: qualitative summaries of precipitation such as frequent rainfall and snowfall ("雨泽频沾"), number of snowy days of three provincial capital cities (i.e., Kunming, Guizhu and Chengdu), snowfall frequency of the three provinces, snowy area (i.e., the number of prefectural-level administrative districts including Fu, Zhou and Ting in which it snowed) and maximum snow

depth in each provinces. (Guizhu was the capital of Guizhou Province during the Qing Dynasty, corresponding to present-day Guiyang.) The snow information in *Yu-Xue-Fen-Cun* in the study area are sometimes incomplete or inaccurate due to the missing months or low spatial precision (aggregated precipitation summary across multiple prefectures). Therefore, we estimated the snow frequency for the Yunnan and Guizhou provinces based on the recorded period, acknowledging some uncertainty, using the following equations:

$$f = \frac{day_r}{Day_r} \tag{1}$$

$$f_{low} = \frac{day_r + 0}{Day_w} \tag{2}$$

$$f_{up} = \frac{day_r + f_{max} \times (Day_w - Day_r)}{Day_w} \tag{3}$$

In equation (1), f is the snow frequency, where $day_r$ is the number of snowy days, and $Day_r$ is the duration of recorded days. In equation (2), $f_{low}$ is the lower bound of frequency, and $Day_w$ is the number of winter days. In equation (3), $f_{up}$ is

the upper bound of frequency, $f_{max}$ is the highest snow frequency observed in *Yu-Xue-Fen-Cun*.

We integrated information from local gazetteers and *Yu-Xue-Fen-Cun* to delineate the southern and western boundaries of snowfall in Yunnan. Based on the precision of the recorded administrative divisions, we extracted the latitude and longitude of the observed snowfall. For county-level observations, we directly used the latitude and longitude of the county, while for prefecture-level observations, we extracted the latitude and longitude of the center of prefecture.

### 3.2 Criteria of temperature indices

The study area covers heterogeneous terrain ranging from basins to mountains and multiple climate zones spans multiple climate zones—from tropical to northern subtropical—resulting in significant intra-regional variability (Fig. 1a). The northern



part of the study area is the Sichuan Basin, with an average elevation of approximately 500 meters, while the southern part encompasses the Yunnan-Guizhou Plateau, averaging around 2,000 meters in elevation. The western part of Sichuan and the

northwest part of Yunnan province were excluded from this study because their climate more closely resembles that of the Tibetan Plateau (Zheng et al., 2013). The cold waves affecting Yunnan province primarily enter from the northeast via the Sichuan Basin or Guizhou province (57%), followed by those entering via the southeast (33.6%), with a smaller proportion from the southeast of Tibetan Plateau (9.4%) (Yang et al., 2023). As these cold waves move southward, the Yunnan-Guizhou Plateau obstructs their path, often causing a quasi-stationary front to form between 103°E and 105°E during winter, resulting

in generally cold and rainy conditions east of the front and milder, sunnier weather to the west (Cai et al., 2022; Duan et al., 2018). Moreover, when cold waves intensify and moves westward and southward, they may further affect the central Yunnan province. Consequently, Guizhou experiences the highest frequency of cold waves and most snowy days within the study area, followed by the Sichuan Basin and central to northern Yunnan Province, while the central and southern parts of Yunnan have the fewest cold wave occurrences (Fig. 1b). It is important to note that winter rainfall and snowfall in the study area are

influenced by multiple factors, including cold air intrusions and water vapor fluxes associated with the India–Burma Trough (Cao et al., 2024; Li & Zhou, 2016).

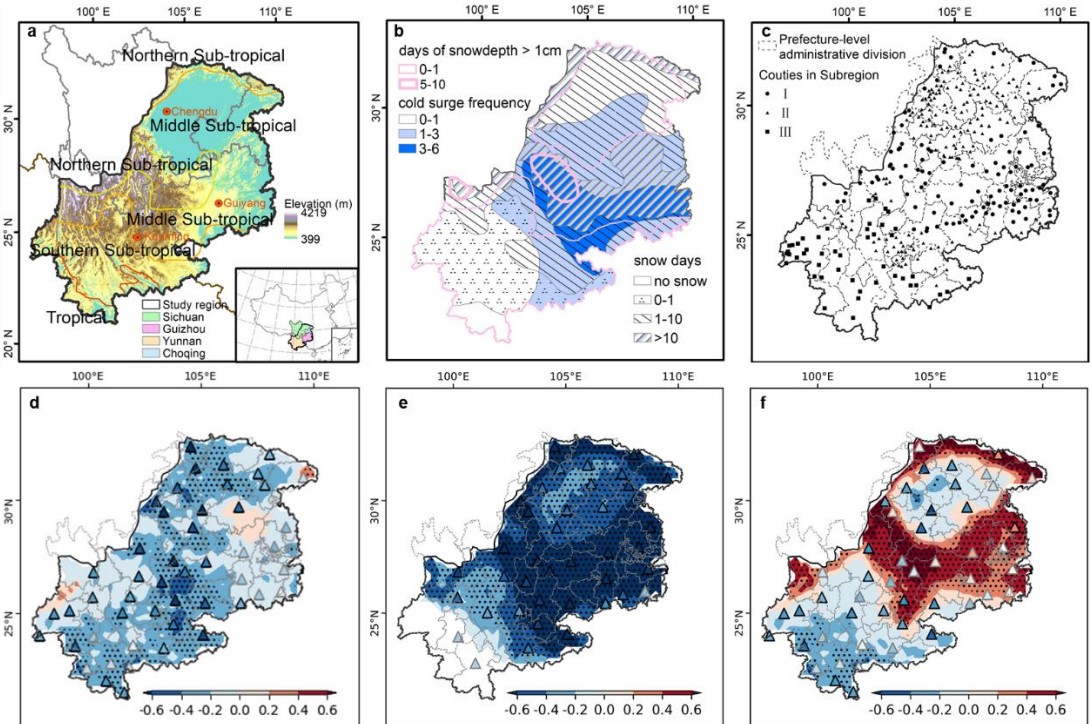

**Fig. 1.** An overview of the study area: a. terrain, climatic zone and capital cities between 1981-2010 (**Zheng et al., 2013**); b. snow and cold surge frequency per year during 1961-2005 (**Chinese Meteorological Administration, 2007**); c. division of

sub-regions; d/e/f. correlation between regional temperature (from CRU) and wet days/snow days/rainy days, contours are derived from DFDP (1961–2015), triangles are derived from GSOD (1942–2019).





Figures 1d, 1e, and 1f specifically illustrate the correlations between regional temperature and the proportions of wet days, snowy days, and rainy days, respectively. Note that the correlation analysis of snowy days was restricted to GSOD stations with snowfall recorded in at least five winters. Regional temperature is negatively correlated with the proportions of wet days

and snowy days across most of the study area. In Guizhou Province, the negative correlation is stronger for snowy days but weaker for wet days (Fig.1e and 1d). However, the correlation between temperature and the proportion of rainy days is less certain. Both GSOD and DFDP dataset show a negative association between regional temperature and rainy days in the Sichuan Basin, as well as in central and southern Yunnan Province (Fig. 1f). It aligns with a local weather proverb in Yunnan province: "No summer or winter in the whole year; once it rains, it is cold as winter (四季无寒暑，一雨便成冬)." Nevertheless, GSOD

and DFDP dataset show inconsistencies in the topographic transition zone between the Sichuan Basin and the Yunnan-Guizhou Plateau. To further discuss uncertainties, we performed correlation analyses using GHCNd and ERA5 hourly precipitation-type dataset (Fig. S3).

Based on the intra-regional climatic variation mentioned above, weather- and climate phenomena or elements in different sites have varied indications of regional temperatures; thus, instead of utilizing the same criteria for the study area, we separated it

into three subregions and applied different criteria. Subregion I includes the northern and middle subtropical zones covering Guizhou, northern Yunnan, and the mountains around the Sichuan Basin with frequent cold surges and more snowy days. Subregion II includes the middle subtropical zone covering the Sichuan Basin and north-central Yunnan. The representative station, Kunming, had snow cover days in only nine winters with a majority only lasting day between 1951 and 2000 (Yang, 2007). Subregion III includes the southern subtropical zone covering central and southern Yunnan with less than one snowy

day per year. We then classified 10 unusual phenomena mentioned in Sec.3.1.1 into different grades (Tab. S1) and assessed their indication in winter temperatures between 1950 and 1999 (Fig. S4). As we expected, the occurrence of a same unusual phenomena in different subregion relates to different winter temperatures. For example, the winters with snow of grade 3 (snowfall lasting 5 or more days) in subregion I were averagely deviated from climatology with around -0.5 standard deviation, which close to -1 standard deviation in subregion II. Finally, combining the features of historical literature, the nature of the

regional climate, and the criteria of the winter cold index for other parts of China in previous studies (L. Ding & Zheng, 2017; Wang, 1990), we established the criteria of the Cold Winter Index (CWI) for Southwest China (Tab. 1).

**Table 1 Criteria of temperature indices**

| CWI | Description | Criteria |
|---|---|---|
| 1 | warm | I/II: flower blooming in winter; <br> or relatively warm winter (Index 2) observed in more than one province; |
| 2 | relatively warm | I: less snow/frozen rain; human perception of warm/hot; <br> or II/III: severe/seasonal drought recorded in one subregion, or moderate/monthly drought recorded in two provinces; |
| 3 | normal | no record; the weather is as normal; |



| 4 | chilly | I: extremely heavy snowfall (depth≥3 chi, 1 *Chi*≈33 cm); snowfall lasting five or more days; three or four separate snowfalls with accumulation; two separate heavy snowfalls; or II: heavy snowfall (depth≥1 *Chi*); snowfall lasting two to four days; two separate snowfalls with accumulation; thin ice in pond or paddy fields; persistent rainy days; or III: snowfall with accumulation; thin ice in pond or paddy fields; human perception of cold; persistent rainy days; |
|---|---|---|
| 5 | cold | I: snowfall lasting ten or more days; five or six separate snowfalls with accumulation; three or four separate heavy snowfalls; thick ice in ponds or paddy fields; death of livestock; or II: snowfall lasting five or more days; three or four separate snowfalls with accumulation; two separate heavy snowfalls; thick ice in pond or paddy fields; completely frozen of water in the containers; freezing rain; death of livestock; or III: heavy snowfall; snowfall lasting two to four days; two separate snowfalls with accumulation; thick ice in ponds or paddy fields; death of livestock; or chilly (index 4) observed in more than one province; |
| 6 | extremely cold | I: persistent snowfall/freezing rain (last longer than one month); seven or more separate snowfalls with accumulation; five or more separate heavy snowfalls; frozen rivers; death of trees/birds/fish/human; or II: snowfall lasting ten or more days; five or six separate snowfalls with accumulation; three or four separate heavy snowfalls; frozen rivers; freezing rain lasting more than ten days; death of trees/birds/fish/human; or III: snowfall lasting five or more days; three or four separate snowfalls with accumulation; two separate heavy snowfalls; freezing rain; frozen rivers; death of trees/birds/fish/human; or cold (index 5) observed in more than one subregion |

## 3.3 Bayesian Reconstruction

### 3.3.1 Prior Generation

First, we calculated the probability distribution of each index for the period 1700–1949, along with the corresponding percentile ranges from cold (index 6) to warm (index 1). Based on these ranges, we defined the temperature intervals corresponding to each index in the simulations. For each winter from a given ensemble member, we assigned an index according to the temperature interval into which it fell. The prior probability for each winter was calculated as the fraction of ensemble members assigned to a given index, relative to the total of 20 members in ModE-Sim. For example, in winter 1700/1701, if 5 of the 20

ModE-Sim members were assigned index 3, the prior probability of index 3 would be 25%. The same procedure was applied using CESM-LME as the prior. Moreover, to assess the reconstruction's dependence on the simulations, we generated a time-independent prior for each winter by randomly selecting 20 from all winters in ModE-Sim (1700-1949), and repeated this procedure 100 times. These three reconstructions—based on ModE-Sim, CESM-LME, and the time-independent ModE-Sim—are referred to as CWI-ModE-Sim, CWI-LME, and CWI-ModE-Clim, respectively.

### 3.3.2 Assignment of likelihood related to temperature indices

Likelihood assignment requires estimating the probability of observations being true given different indices. Previous studies generated subjective estimates of likelihood based on expert knowledge from historians (Camenisch et al., 2022). In this study, we advance the methodology by combining subjective estimates with objective observations to estimate likelihood, thereby





providing greater flexibility when priors may be biased as mentioned in Sec. 2.2. First, we assigned each winter between 1950

and 1999 an index based on the simulations. For example, if index 6 corresponds to the coldest 5% of winters, we calculated

the corresponding temperature threshold in the simulations and assigned winters from 1950–1999 falling within this range to

index 6. For winters of each index, we then calculated the frequency of different phenomena recorded in literatures between

1950 and 1999, as a basis for assessing the dependence of observations on climate. We assume that observations during this

period carry relatively lower uncertainty, with fewer missing values and recording errors. Based on this, the final likelihood

of each winter during 1700–1949 was adjusted to incorporate uncertainties in the documentary data, including source reliability,

record quantity, and descriptive accuracy. Examples and results are presented in Section 4.2.1.

### 3.3.3 Posterior calculation and temperature reconstruction

The information from documentary data and the simulations were converted into probabilities, which were multiplied and

subsequently normalized to obtain posterior probabilities, as defined in Eq. (4).:

$$p(T|evidence) = p(T) \times \frac{p(evidence|T)}{p(evidence)} \tag{4}$$

where p(T) is the prior probability from paleo-simulations and p(evidence | T) is the likelihood based on documentary data,

while p(T | evidence) is the posterior probability. Note that the term on the right is divided by a normalizing constant, ensuring

that the posterior probabilities form a valid distribution summing to 1.

Finally, winter temperatures were reconstructed as a weighted average of the mean values associated with each index, using

posterior probability distributions as weights, as given in Eq. (5):

$$T_{rec} = \sum p(T|index = n) \times \overline{T_n} \tag{5}$$

where $T_{rec}$ is the reconstructed winter temperature of southwest China, n is the temperature index, p(T | index=n) is the posterior

probability conditional on index n, and $\overline{T_n}$ is the mean value of temperatures associated with index n. Moreover, to illustrate

the confidence in our reconstruction, the mean temperature associated with each index was replaced by 10th, the 25th, the 75th

and the 90th percentile. This approach allows us to present the confidence interval of possible temperature estimates rather than

a single value, thereby providing the reconstruction uncertainty.





## 4 Results

### 4.1 From weather and climate-related information to cold winter index

### 4.1.1 Weather and climate-related information recorded as anomalies and systematic reports

Based on information in local gazetteers, Figure 2a shows the cumulative observed frequency of ten categories of unusual phenomena from 1700 to 1999, representing the proportion of years in which each phenomenon had been recorded from 1700 up to each given year. Figures 2b and 2c show the fractions of each phenomenon during 1700–1949 (historical era) and 1950–1999 (recent era), illustrating the characteristics of observations in the two periods. Observations of snowfall dominated both eras, with the proportion rising from 0.2 to 0.4 between the 1730s and 1890s and remaining around 0.4 throughout the twentieth

century. Similarly, precipitation anomalies (mostly drought) were recorded in about 20% of years before 1850, increasing to 30% thereafter. However, other phenomena showed greater differences between the two eras. For example, freezing rain, a common winter phenomenon in Guizhou Province, was rarely considered abnormal and recorded in the historical era, with only severe events noticed and documented, which could contribute to index reconstruction, whereas observations recordrd increased markedly in the recent era. The impacts of cold events on biological systems were also more frequently recorded in

recent years, due to more complete observations and the larger scale of population and agriculture, which increased exposure to cold surges. Specifically, historical impacts were primarily limited to dead trees (e.g., banyan trees) and animals (e.g., birds, fish, livestock), while recent impacts mainly involved damaged or dead crops and livestock.

Regarding winter flowering, which is one of the few phenomena related to warm winters, local gazetteers have not recorded it as abnormal since 1950. Because the end of the lunar year during 1700–1949 corresponded to no later than February 18 in the

Gregorian calendar, we interpret flowering events occurring before this date observed after 1950 as equivalent to "winter flowering" in narrative sources. And three events are found in the *Phenology Report*: apricot (*Prunus armeniaca* L.) at Renshou (30°N, 104°E, 430 m, Subregion II) reached peak flowering on February 16, 1965, and February 10, 1966; and in Guiyang (27°N, 107°E, 1,050 m, Subregion I), willow (*Salix babylonica* L.), apricot (*Prunus armeniaca* L.), and Plum (*Prunus persica* (L.) Batsch) reached peak flowering before February 18, 1987. Overall, local gazetteers in the historical era tended to "record

the extraordinary rather than the ordinary," documenting events that deviated from the norm, whether unusually cold or warm. In contrast, literatures in the recent era primarily focus on disasters, emphasizing events that affected daily life, agriculture, and other aspects of people's livelihoods, including some that were not truly extraordinary. Snowfall and precipitation anomalies are more comparable between the two periods, likely because such events are easier to be noticed and have great impacts on agriculture, making them less likely to be overlooked.








**Fig. 2. Unusual phenomena: (a) the cumulative occurrence proportions of 10 categories of unusual phenomena documented in historical records; the outside rings in (b) and (c) are the percentage of observed phenomena during 1700-1949 and 1949-1999 (normalized), the inner rings are the regional results in which the dark, medium and light colors show subregion I, II, and III respectively.**

Based on the *Yu-Xue-Fen-Cun* records, we distinguished between snow that accumulated on the ground and snow that did not

accumulate (light snow or snow melting upon falling) in Guizhu and Kunming (Fig. 3a, b). Before 1911, Guizhu experienced

six winters with more than seven snowfall days, five of which occurred in the nineteenth century and included over seven days

with snow accumulation, with a maximum of 13 snowy days recorded in the winter of 1816/1817. According to the GOSD,

Longdongbao Station—the closest station to Guiyang—recorded its maximum number of snowfall days without rainfall in the

winter of 1976/1977, one of the coldest winters in southwest China during 1950–1999, with a total of seven snowfall days. In





Kunming, the contrast in the number of snowy days is more evident between the eighteenth–nineteenth centuries and recent decades. According to Yang (2007), nine winters with snowfall with accumulation in Kunming and its surrounding areas were observed between 1951 and 2000, though most events lasted only one day; the heaviest snowfall in the winter of 1982/1983 lasted two and three days, respectively. It should be noted that the number of snowy days, while partly reflecting temperature

variability, is also strongly influenced by moisture transport, and thus cannot fully explain temperature changes at a single station or across the region.

The southernmost and westernmost snowfall boundaries in Yunnan Province (Fig. 3d) help infer the intensity and spatial extent of cold surges. Variations between the southern and western boundaries show a high degree of consistency, corresponding to the primary northeast-to-southwest pathways of cold wave incursions. Before 1950, the eastern boundary of snowfall reached

east of 100°E in 31 winters and south of the Tropic of Cancer in 7 winters, compared with 12 and 4 winters, respectively, during 1950–1999. The southern boundary of snowfall reached 22.78°N in winter 1983/1984, similar to the winters of 1782/1783 and 1797/1798, indicating that the cold wave reached the southernmost part of Yunnan and affected the entire province. It should be noted that the observations represent only the southernmost recorded snowfall locations; the actual southernmost extent may have been further south, though this uncertainty has substantially decreased in recent decades due to

more comprehensive observations.

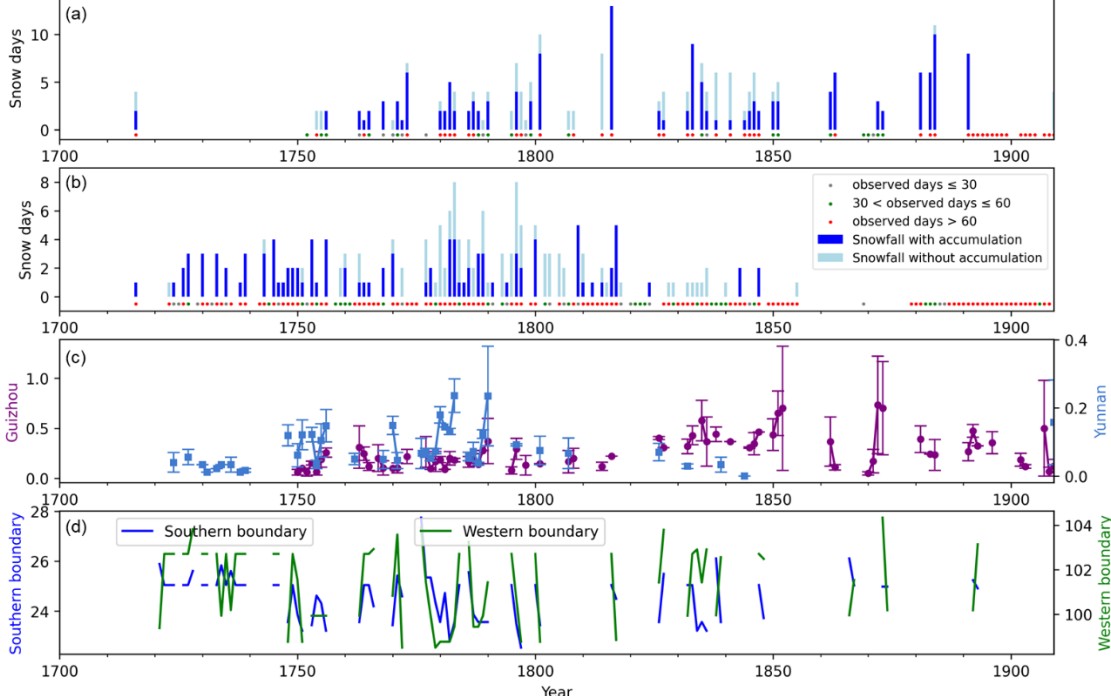

**Fig. 3. Snow information: (a) Number of snowy days in Guizhu; (b) Number of snowy days in Kunming (colored dots indicate winters with observations, while missing dots indicate no observations); (c) Snow frequency estimates for Guizhou and Yunnan provinces; (d) Southern (°N) and western (°E) boundaries of observed snowfall.**





It is also crucial to discuss the accuracy and completeness of weather and climate information from different sources for estimating the likelihood. Ge & Zhang (1990) proposed a framework for evaluating weather and climate information in historical literatures along three dimensions—time, space, and the description. In this framework, the closer the observer is in time and space to the climatic event, the more detailed the record is, the more accurate and reliable the record is. For example, if the climatic information and the observer's location are in the same county, prefecture, or province, the reliability of the

record is estimated to be 0.9–1.0, 0.8–0.9, and 0.7–0.8, respectively. In general, official archives are regarded as the most accurate, followed by private notes, and finally local gazetteers. Yang (2006) also discussed the reliability and systematic bias in the *Yu-Xue-Fen-Cun* archive in Yunnan Province. The archive is found to be very sensitive and reliable for snow events, but less sensitive for drought events. Moreover, the richness of information in the archives gradually declined with the different reigns. The information was the richest in the archive during the Yongzheng and Qianlong reigns, followed by the Jiaqing and

Daoguang reigns, while it was the most vague and general during the Guangxu reign. We also found that the archives of Guizhou Province exhibit similar changes. As shown in Figure 3(a), during the late Guangxu reign, some archive from Guizhu mention rainfall but no snowfall, which contrasts with the climatic features of earlier periods, modern observations, and broader regional patterns. Consequently, the reliability of *Yu-Xe-Fen-Cun* appears to have declined over time.

### 4.1.2 Cold winter index reconstruction

The cold winter indices (CWI) from 1700 to 1950 in this study are moderately correlated (spearman correlation = 0.54, p<0.01) with the winter temperature grades derived from instrumental data (Institute of Meteorological Sciences & Central Meteorological Observatory, 1984) between 1910-1949 (Fig. 4a). The largest discrepancy between the two indices was observed in the consecutive winters of 1928/1929 and 1929/1930. The winter of 1929/1930, in particular, was deemed extremely cold due to severe freezing rain in central Guizhou, described in historical records as 'freezing not seen in past forty

years', and the frozen rivers with dead fish in Sichuan Province. However, the instrumental-based index is 2.87 for this winter, indicating a normal winter. To further discuss the differences, the DJF temperatures from three representative stations in the early 20th century—Guiyang (26.58°N, 106.72°E), Kunming (25.12°N, 102.9°E), and Chongqing (29.55°N, 106.55°E)—were extracted and analysed (Fig. 4c) (Central Meteorological Bureau & Institute of Geophysics, Academia Sinica, 1954). We found that, although January 1930 was very cold, consistent with historical records, the rapid warming in February resulted in the

DJF mean temperature being relatively mild. This also illustrates the uncertainty of the *index* method based on descriptions about abnormal events, particularly when substantial intra-seasonal variations occur.



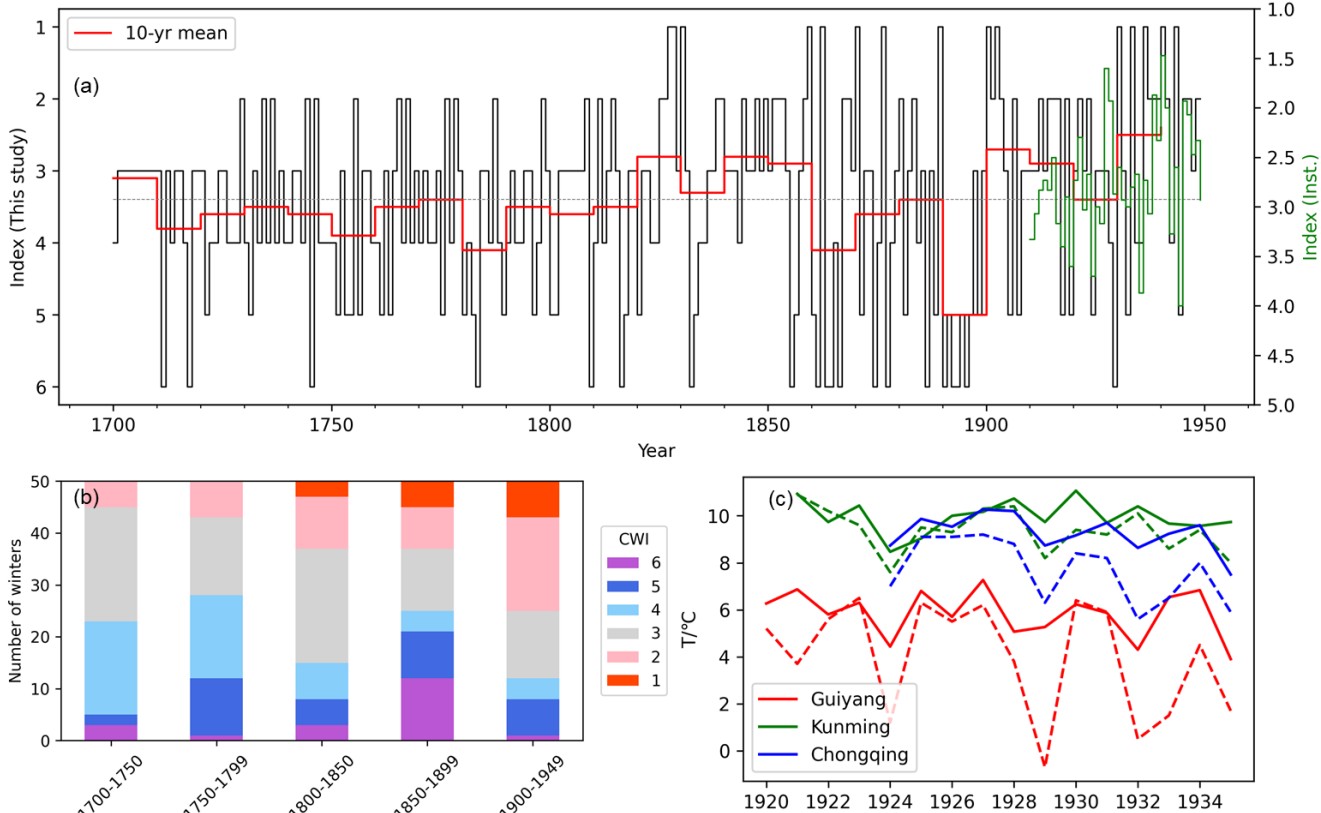

**Fig. 4. Cold winter indices between 1700-1949: (a) cold winter indices; (b)distributions of CWI in every half century; (c)instrumental of presentative stations in the early 20th century (solid lines are DJF mean temperatures, dashed lines are January temperatures)**

We identified 20 extremely cold winters, 34 cold winters, 49 chilly winters, 48 relatively warm winters, 15 warm winters, and 84 normal winters. The winters of 1809/1810, 1855/1856, 1861/1862, 1871/1872, 1877/1878, 1886/1887, 1892/1893, 1917/1918, and 1929/1930 were identified as cold winters both in this study and in southeast China demonstrating the inter-regional linkages of winter climate under the dominance of East Asia Winter Monsoon (EAWM) (Hao et al., 2011). In terms of temporal distribution, there were five cold and extremely cold winters in the first half of the 18th century; twelve in the

second half of the 18th century; eight in the first half of the 19th century; 21 in the second half of the 19th century; and eight in the first half of the 20th century. Additionally, the frequency of warm winters was also higher in the 19th century than in the 18th century, indicating a period of pronounced interannual variability (Fig. 4b).

On decadal scales, winter temperature variations since 1700 can be divided into four phases: a relatively stable and cold phase between the 1700s and 1810s, a warm phase between the 1820s and 1850s, a cold and variable phase between the 1860s and

1890s, and a warming phase from the 1900s onward. Cold winters often occurred in clusters, such as 1860–1863 and 1890–1896, making the 1890s the coldest decade in the past 250 years. Compared with the decadal temperature reconstruction of Kunming (Yang, 2007), our results show a similar 4 phases of cold-warm-cold-warm change, with some differences in the



18th century. In Yang's study, the first half of the 18th century continued the coldness of the 17th century, with coldness comparable to the second half of the 19th century, while it entered a stable period similar to that of the second half of the 20th
century after the rapid warming in the 1760s. In contrast, in the CWI series of this study, the first half of the 18th century is similar to the second half, although the warming in the 1760s is evident.

## 4.2 Winter temperature reconstruction between 1750-1949

### 4.2.1 Likelihood estimation

Our estimation of likelihood consists of two components: the reliability of the observations and the dependence of these
observations on the climate. The former, as discussed in Section 4.1.1, is evaluated based on literature sources, the number, consistency, and spatial coverage of the records, as well as the potential for omissions. The latter will be examined using observational records from 1950 to 1999.

We calculated the temperature ranges corresponding to each cold winter index in ModE-Sim and CESM-LME based on the quantiles associated with each index, and used these ranges to assign indices to winters in 1950–1999 based on CRU
temperature data (Fig. 5a). Compared with CRU (1950–1999), the temperature distributions of the two simulation ensembles for 1700–1949 are shifted toward colder and warmer values, respectively. As mentioned in Section 2.2, these two ensembles show large biases at low latitudes, whereas the temperature distributions for 1950–1999 are very similar. Accordingly, the number of winters assigned to indices 1–6 in 1950–1999 was 8, 16, 19, 4, 3, and 0 for ModE-Sim, and 2, 2, 13, 12, 12, and 9 for CESM-LME. The temperature threshold corresponding to index 6 in ModE-Sim (reference period 1950–1999) is $-1.68\,°C$,
which is lower than the temperature anomaly of the coldest winter of 1967/1968 during this period. Therefore, no winter between 1950 and 1999 was classified as index 6. We then calculated the frequency of different phenomena occurred in winters corresponding to index from 1 to 6, and the results based on the two simulation ensembles were as significantly different as expected (Fig. 5b) which could provide a baseline for the likelihood estimation when using different simulations as prior. Taking heavy snow and severe drought in subregion II as an example, the frequency of observed heavy snow in normal winters
(CWI=3) based on ModE-Sim is around 24% which is about 28% based on CESM-LME, and the frequency of observed drought in normal winters based on ModE-Sim is 19% which is about 16% based on CESM-LME. This is because the winter temperatures in ModE-Sim and CESM-LME during 1700-1949 shifted towards colder and warmer conditions, respectively, relative to the instrumentals (CRU) during 1950-1999. Therefore, the same temperature may be classified as index 3 in ModE-Sim but as index 4 in CESM-LME.



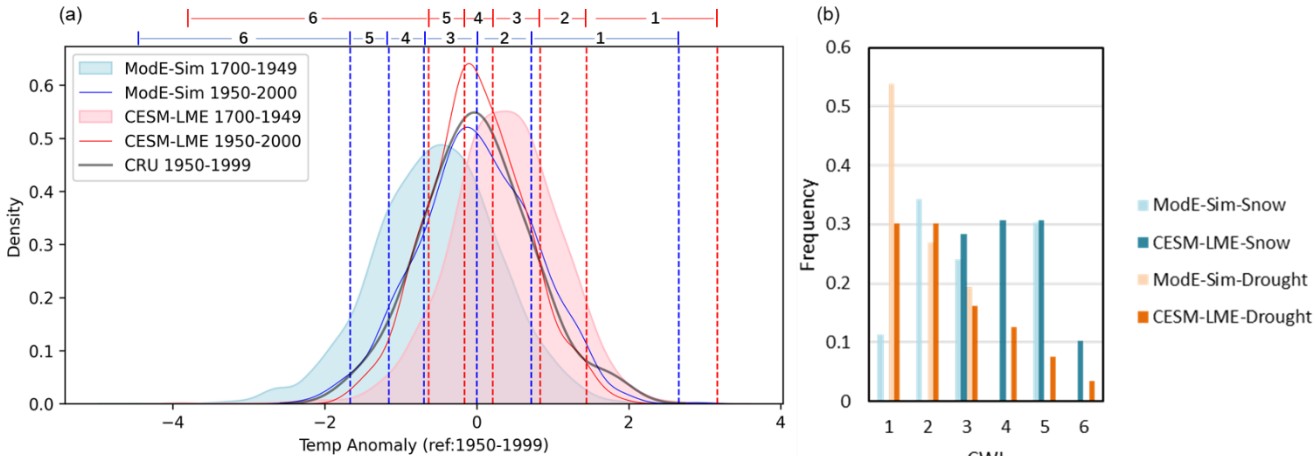


**Fig. 5 (a) Probability density of simulations (1700-1949) and CRU (1950-1999), the vertical dashed lines indicate the temperature intervals corresponding to each index; (b) probability of phenomena (heavy snowfall and severe drought in zone II) within different indices;**

Based on the climate dependence of the observations discussed above, the period from 1950 to 1999 was taken as a reference

to estimate the likelihood for each winter from 1700 to 1949. Subsequently, the likelihood was adjusted according to the reliability and accuracy of the historical records. To demonstrate our likelihood estimation process, we take the winter of 1870/1871 as an example, and show some typical winters corresponding to different CWI as examples based on ModE-Sim (Tab. 2). The winter of 1870/1871 was assigned with an index of 1. Evidence from various sources, including local gazetteers and the *Yu-Xue-Fen-Cun* records, consistently indicates signs of warmth extending over a wide spatial region and spanned the

whole winter with detailed descriptions, suggesting that the observations for this winter are highly reliable. Then we identified winters during 1950-1999 with similar observations. Based on the ModE-Sim, after normalization, such observation occurred in warm winters with a frequency of approximately 61%, in relatively warm winters with 26%, in normal winters with 13%, and in cold winters with 0 which is under the assumption that the observations are reliable. There is also a small probability that the observations for this winter are unreliable or that certain phenomena were not recorded. In this case, after normalization

and assigning a small portion of probabilities to cold winters, the final estimated likelihoods are 58%, 25%, 14%, 1%,1%, and 1% for CWI levels 1 to 6, respectively.

**Table 2 Examples of Likelihood estimation**

| Winter | CWI | Summary | p(evidence \| T) | | | | | |
|--------|-----|---------|---|---|---|---|---|---|
| | | | **1** | **2** | **3** | **4** | **5** | **6** |
| 1870/1871 | 1 | In Guizhou province, only three the snowed prefectures were fewer than normal winters, and no snowfall mentioned in Guizhu. The Sichuan and Yunnan Province both suffers from a prolonged drought in winter. In Xuanwei, a city in northeast Yunnan, there was no snow or freezing in this winter. | 58 | 25 | 14 | 1 | 1 | 1 |



| 1867/1868 | 2 | Winter in Xingyi county (in Guizhou) was snow-free, southern Sichuan remained dry through winter and spring, and central Yunnan experienced heavy snowfall during November to January. | 32 | 38 | 16 | 12 | 1 | 1 |
|---|---|---|---|---|---|---|---|---|
| 1701/1702 | 3 | with no data; | 15 | 27 | 27 | 20 | 10 | 1 |
| 1722/1723 | 4 | Kunming in central Yunnan experienced heavy snowfall in January 1723; No records in Guizhou and Sichuan Province. | 2 | 15 | 18 | 18 | 42 | 5 |
| 1761/1762 | 5 | In southern Sichuan Basin, two heavy snowfalls occurred in January, on 4–5 January 1762 and 18–20 January 1762. The precipitation of Yunnan Province is normal in January based on the Yu-Xue-Fen-Cun. | 1 | 6 | 20 | 30 | 41 | 2 |
| 1892/1893 | 6 | In Guizhou Province, heavy snowfall occurred in January, the extreme cold caused rivers to freeze, allowing people to walk on the ice, and led to widespread fish mortality. In Sichuan Province, rivers and paddy fields froze, and the severe cold resulted in the death of egrets and banyan trees. Central and northeast Yunnan Province also experienced exceptional snowfall during this winter. | 1 | 2 | 3 | 4 | 20 | 70 |

### 4.2.2 Winter temperature reconstruction

As described in Section 3.3.3, we calculated the posterior probability by multiplying the prior by the likelihood and normalizing
it. Using the posterior, we reconstructed winter temperatures since 1700 and estimated their uncertainties by weighting quartiles of temperatures corresponding to different CWI (Fig. 6).

The temperature series shows high interannual variability, with the coldest winters being 1783/1784, 1816/1817 and 1892/1893, nearly 2°C colder than the reference period, and the warmest winter being 1827/1828 but not exceeding the 20th century.

Compared with Yang's reconstruction, the amplitude of our sequence is slightly smaller which might because the difference
between regional and local scale. In the 18th century, our reconstruction results difference about the warming in the 1760s.Although our reconstruction also shows a brief warming period in the 1760s, it did not enter a more stable warm period like Yang's reconstruction, but instead experienced a rapid decline. The reasons may stem from two aspects: firstly, Yang's reconstruction considered only snowfall and precipitation data from Kunming and its surrounding areas, whereas our study incorporated snowfall in the whole province; secondly, the *Yu-Xue-Fen-Cun* archive utilised by Yang were incomplete, which
we supplemented with sources from the Qing Dynasty Archives Retrieval System of the National Palace Museum, Taipei. Here we take the winter of 1763/1764 as an example. The archives used by Mr Yang made no mention of snowfall in Kunming during that winter; however, the records we discovered not only referenced snowfall in Kunming in January 1764, but also documented snowfall occurring across multiple prefectures in central and western Yunnan Province (https://qingarchives.npm.edu.tw/index.php?act=Display/image/1184792tMlPWRg#95l). Among these, the snow depth in
Menghua Fu reached one to two *Chi*. Consequently, this winter remains classified as a cold winter in this study. Ultimately, the correlation between our reconstruction and Yang's reconstruction increased from 0.15 to 0.49 compared to ModE-Sim. Although the historical documents we used overlap to some extent, the results still demonstrate the effectiveness of the Bayesian approach.




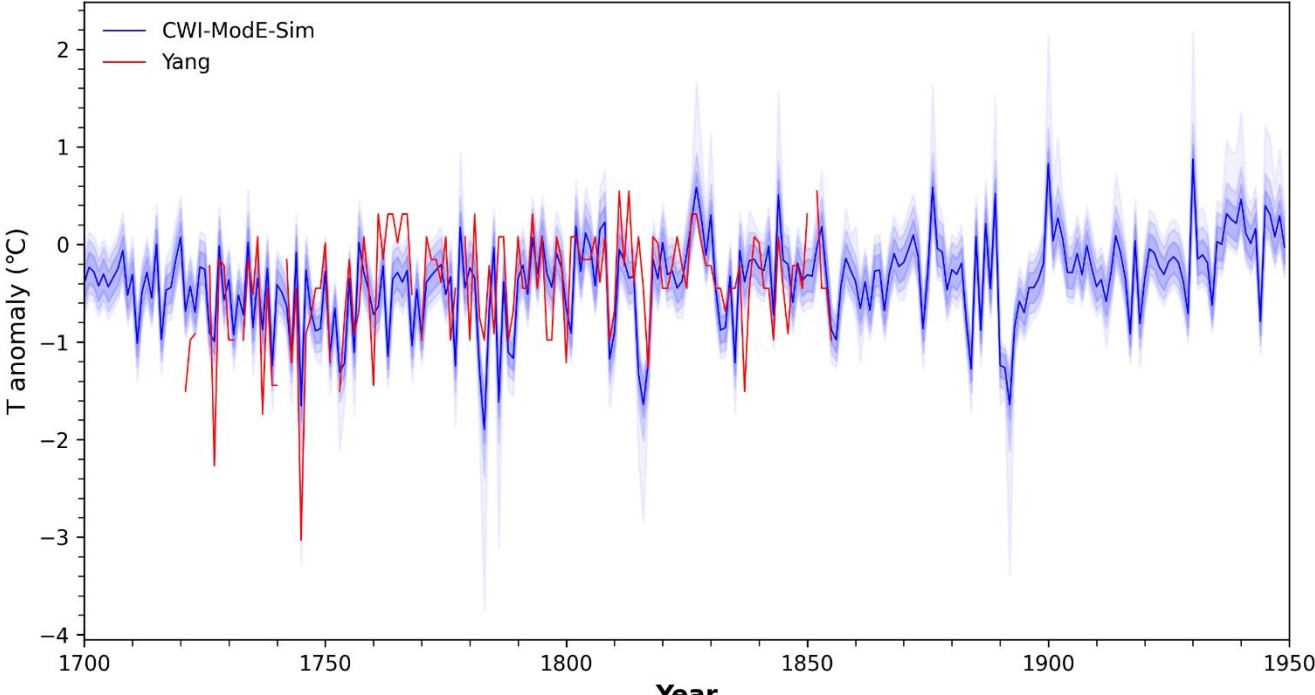

**Fig. 6 CWI-ModE-Sim and Yang's reconstruction (ref: 1950-1999; for Yang's reconstruction, the reference is extracted of Kunming station in GHCNd dataset)**

Our results showed the bias entered from these two prior we employed were significantly reduced after reconstruction, with the correlation improving from 0.15 to 0.60. We also compared our reconstruction with the independent assimilation WinTEDA (Steiger & Tejedor, 2022), and the results showed that the correlation was also improved to some extent (Tab. 3).

**Table 3 Correlation before and after assimilation (* $p \leq 0.05$,** $\leq 0.01$ )**

|        | ModE-Sim with CESM-LME | ModE-Sim with WinTEDA |
|--------|------------------------|------------------------|
| before | 0.15*                  | 0.16*                  |
| after  | 0.60**                 | 0.19*                  |

### 4.3 Source of information: observations and simulations

The CWI-ModE-Clim was reconstructed using a time-independent prior, and the comparison among CWI-ModE-Clim and ModE-Sim is shown in Fig. 7. The simulations and the observations provide broadly consistent information on some cold winters, especially those following major volcanic eruptions—such as the winters of 1783/1784, 1808/1809, and 1817/1818— although the simulations generally show a stronger cooling than the historical records. However, for some winters, the simulations and the documentary evidence offer contradictory signals. For example, the winter of 1870/1871, identified as a warm winter in the documentary sources, appears colder than the climatology in the simulations. As for the warm period of





the 1820s, the simulations and the historical documents are again in agreement, suggesting that the climate model may have successfully captured the underlying mechanisms associated with warm winters.

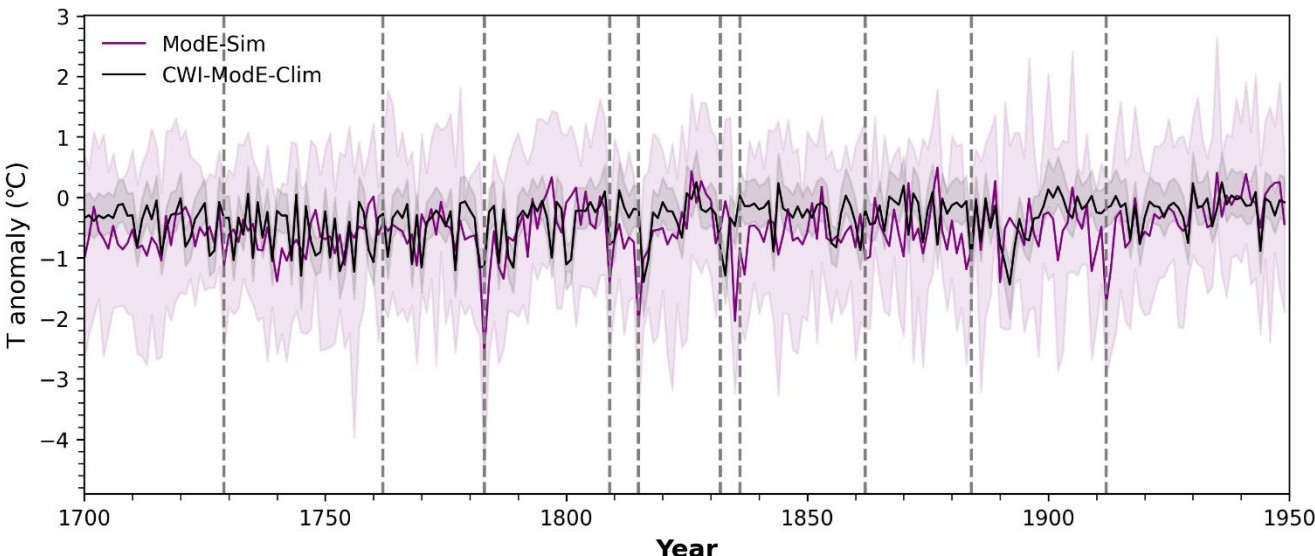


**Fig. 7 CWI-ModE-Clime and ModE-Sim,** the vertical dashed lines denote the 10 biggest eruptions during 1700-1949 (Sigl et al., 2015)

To further discuss the uncertainty from simulations, we compare the atmospheric circulation pattern related to the cold/warm winters in the reanalysis (i.e. ERA5) and paleo-simulations (i.e. ModE-Sim) to discuss the climate model's ability to capture the climate variability. We defined a winter as cold or warm if the DJF temperature deviates from the climatology by 1σ. For

ModE-Sim, cold/warm winters were identified separately within each member, after which a composite analysis was performed using all the identified cold/warm winters. Compared with warm winters, cold winters are characterized by a pronounced Siberian High in both reanalysis and paleo-simulations, extending over most regions of China and inducing anomalous northeasterly or easterly winds in eastern and southwestern China. The cold air advected from the north is a major factor contributing to the temperature reduction (Figures 7a and 7d). In the mid-troposphere, the geopotential height exhibits

a meridional pattern, with a positive anomaly over the Siberian region. Additionally, the Western Pacific Subtropical High (WPSH) is significantly intensified during warm winters in ERA5. Similar differences are observed in paleo-simulations, although the intensity of the WPSH are much less reproduced (Figures 7b and 7e). Notable differences also appear in the composition of westerly between cold and warm winters. In the upper troposphere, the 200 hPa zonal wind presented a tripolar pattern, with a pronounced intensification of the subtropical westerly jet during cold winters. Overall, the circulation patterns

across the lower, middle, and upper troposphere associated with winter temperatures over southwest China in paleo-simulations are consistent with reanalysis and previous studies(Jiang & Li, 2010; Shen et al., 2015; Shu et al., 2021), indicating that climate models can effectively capture the regional temperature configurations in relation to large-scale circulation.



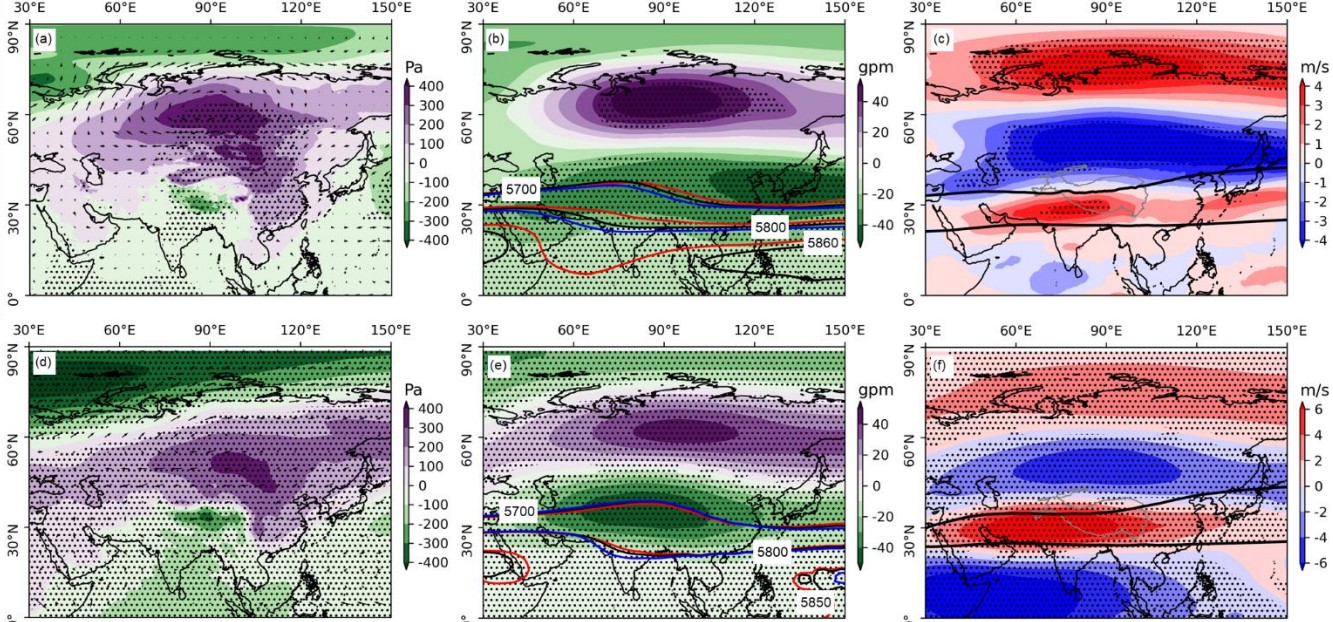

**Fig. 7. Atmospheric circulation composition** (cold winters minus warm winters) (a) sea level pressure and winds on 850hPa level, (b) geopotential height on 500 hPa (black for climatology, blue for cold winters mean while red for warm winters mean) and (c) zonal wind on 200 hPa (colorful shading) and jet (wind speed $\geq$ 40m/s) for climatology in ERA5 during 1950-2019; (d~f) same variables in ModE-Sim during 1700-1949.

## 5 Discussion

Climate reconstruction has developed numerous rigorous methodologies and achieved substantial progress at the global scale; however, regional and seasonal coverage remain uneven. In this study, we employed an enhanced Bayesian approach to reconstruct a 250-year winter temperature series in Southwest China, characterized by complex natural environments and scarce data. The reconstructed winter temperature variations over the past three centuries are consistent with previous studies, confirming the occurrence of distinct warm and cold periods, while also demonstrating the feasibility and flexibility of the Bayesian framework. This study used two paleoclimate simulations with considerable biases as priors, and the substantially reduced biases in the reconstructions indicate that the documentary data were effectively assimilated.

In terms of methodology, the Bayesian approach based on historical documents has the advantage of incorporating all available information; however, it is often challenged in terms of subjectivity and robustness when compared with traditional methods. To address these issues, we improve upon previous approaches by using the recent observations to present the dependence of observations on climate. This not only reduces subjectivity but also allows adaptability to different prior simulations. Our aim is to advance the methodology of historical climate reconstruction by providing a probability distribution–based perspective that maximizes the use of the historian's expertise. The asymmetric confidence intervals in our reconstruction further




demonstrate the Bayesian method's ability to integrate uncertainties from documentary sources into the final results, and to derive estimates even in the absence of direct evidence—a feature distinct from traditional optimal estimation methods. Moreover, as an assimilation method, the Bayesian approach enables identification of the information sources in the 455 reconstruction results and shows considerable potential for analysing the forcing of extreme events.

However, the Bayesian approach used in this study still has certain limitations. On one hand, although basing the reconstruction on indices reduces the data requirements compared with other methods, it remains sensitive to the quantity and quality of observations in both historical and recent periods. The index is more effective at characterizing extreme events but less effective at capturing lower-frequency variations. The distinction between literature in the historical era and modern era may 460 also introduce uncertainty. On the other hand, although we have provided a baseline using recent observations, due to the complexity of the information in the literature data, different researchers may provide different opinions on the likelihood estimates Therefore, our goal is to enhance the robustness of the method as much as possible.

## 6 Conclusions

This study extracted abnormal phenomena recorded in local gazetteers and snowfall information from the *Yu-Xue-Fen-Cun* 465 archive to generate the Cold Winter Index (CWI) for Southwest China from 1700 to 1949. Using a Bayesian approach, we integrated the climate information from historical literature, along with its associated uncertainties, into paleoclimate simulations to reconstruct winter temperatures in Southwest China. The respective contributions of simulation and documentary data were also discussed.

The results show:(1) From 1700 to 1949, Southwest China experienced a total of 20 extremely cold winters and 34 cold winters. 470 Among these, nine winters coincided with cold winters in eastern China, indicating that cold waves along the northeast–southwest route created significant interregional connections. On the decadal scale, the 1890s represented the coldest decade over the past three centuries. On a multi-decadal scale, the second half of the 19th century exhibited the highest frequency of extreme cold winters and relatively strong interannual variability. (2) The final reconstruction is consistent with the previous reconstruction on the decadal and multidecadal scales, with an amplitude of ~2.77 °C. The paleo-simulations can correctly 475 capture the atmospheric circulation patterns associated with extreme cold and warm winters, and has significant contributions to the reconstruction especially in the years following large volcanic eruptions.

This study demonstrates the feasibility and flexibility of the Bayesian approach in the southwest China, which has the potential to incorporate information of different types and degrees of accuracy from the literature into the reconstruction, and effectively reduce the influence of bias from the prior.



**Data availability statement:** The ModE-RA, ModE-RAclim, and ModE-Sim data (Valler et al., 2024) can be downloaded from DKRZ (https://www.wdc-climate.de/ui/entry?acronym=ModE-RA). CESM-LME are available from the NCAR Climate Data Gateway (https://www.earthsystemgrid.org). The CRU TS 4.07v can be downloaded from CRU (https://crudata.uea.ac.uk/cru/data/hrg/cru_ts_4.07/). The Global Surface Summary of the Day (GSOD) dataset is available from the NCEI (https://www.ncei.noaa.gov/data/global-summary-of-the-day/). The DFDP are available from the National

Tibetan Plateau Data Center (https://data.tpdc.ac.cn/zh-hans/data/fcb2e6a0-12a8-4607-aff1-d870bedb9056). The CWI, prior, likelihood, posterior and reconstructions based on the two ensembles can be downloaded from the figshare doi.org/10.6084/m9.figshare.30674369

**Code availability statement:** All calculation and analyses were done Python using standard code.

**Author contributions:** SC performed the reconstruction and analysis, SB supervised the work and provided overall guidance.
All authors contributed to writing the paper.

**Funding Information:** The work was funded by the China Scholarship Council, grant No. 202206040033; Swiss National Science Foundation, grant No. 219746.

**Competing interests.** The contact author has declared that none of the authors has any competing interests.

**Acknowledgements.** I would like to thank Yuda Yang for providing the *Yu-Xue-Fen-Cun* archive.

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
