# Peer review of "Reconstruction of winter temperature of southwest China over the past 300 years based on a Bayesian approach"

_EGUsphere, 2025_

## Referee Comment (RC1)

Review "Reconstruction of winter temperature of Southwest China"

This manuscript provides an important new reconstruction of winter temperature for southwestern China as well as an important demonstration of, and update to, the Bayesian historical climatology approach introduced in Camenisch et al (2022). The new methods and results of this manuscript look solid. Nevertheless, the presentation needs some improvements. I was already familiar with the Bayesian approach before I read this manuscript, yet I still found it difficult to understand some of its methods and results. I suspect the current manuscript would be confusing or challenging for many readers. Moreover, the manuscript could do a better job presenting both advantages and challenges of the Bayesian approach. Therefore, I recommend publication after revisions. The following review highlights places in the manuscript that need particular attention.

Line 22: The qualification "data-sparse" is relative. It would be better to explain that there is abundant data for eastern China but relatively less for southwestern China.

L35: Should this be high spatial *resolution*? In most parts of the world, the spatial *coverage* of documentary data is usually not very good, with most observations concentrated in cities or a few rural locations. Is this different in SW China?

L38: It seems the manuscript uses "climatology" here and throughout the paper to mean "average weather" or "long-term averages." It would help to specify the meaning of the term as it used here.

L40-50: The paper should expand and clarify this discussion of the strengths and limitations of conventional index methods and the possibilities of the Bayesian method, so that readers can better understand what is at stake. Some specific points to note:
-The reconstruction skill for the conventional index methods is usually best for decadal-scale variability (as discussed in Rudolf Brázdil et al., "Historical Climatology in Europe - The State of the Art," *Climatic Change* 70 (2005): 363–430 and Christian Pfister et al., "Analysis and Interpretation: Temperature and Precipitation Indices," in *The Palgrave Handbook of Climate History*, ed. Sam White et al. (Palgrave Macmillan UK, 2018).)
-Conventional index methods do not combine historical climatology and paleoclimatology. They use only archives of society.
-Conventional index methods must leave gaps in a reconstruction where there is no suitable documentary evidence.
-Reconstructions based on conventional index methods make one best estimate of the value of the target variable rather than a probability distribution. The historical sciences are starting to recognize that these probability distributions are important for understanding (environmental) history. (See e.g., Myles Lavan, "Epistemic Uncertainty, Subjective Probability, and Ancient History," *Journal of Interdisciplinary History* 50 (2019): 91–111.)
-Other methods may also incorporate both documentary information and paleoclimate proxies. The paper should also compare this Bayesian methods to past methods for integrating historical climatology and paleoclimatology together into temperature reconstructions (e.g., Jürg Luterbacher et al., "European Seasonal and Annual

Temperature Variability, Trends, and Extremes Since 1500," *Science* 303 (2004): 1499–503). This may require a little more explanation of how the Bayesian method gets more information out of the sources than a single index value.

L55-59: I would recommend rewriting these lines for clarity and precision. First, it creates a lot of confusion to include the CWI as part of the "data" rather than the reconstruction "method." The method in this study was to start by creating a winter temperature index, then to analyze those index results, and finally to use the index together with an ensemble of simulations to produce a reconstruction (i.e., posterior probability distribution). By treating the CWI as "data" it seems to be something that was "given" (the literal meaning of "data") rather than something created through analysis of records. In fact, even the term "documentary data" may be misleading here. In this Bayesian approach, you are using the presence or absence of different documentary records as a means to update your beliefs about past weather and climate. Therefore, I would just say "documentary records." Moreover, I wouldn't say that the records "contain information about winter climate or meteorological events." To maintain the strict logic of the Bayesian method, you should say that "winter climate and meteorological events shaped the contents of these records." Causation always goes from the weather to the documents. The phrase "expressed through cold winter indices" is also confusing, since it suggests that the documents themselves included these indices.
I think I understand what you mean by the sentence "Paleo-simulations provided a climate background where all the information into which all information was assimilated." However, the sentence literally makes no sense and it needs to be rewritten. The meaning of the following sentence ("The instrumental data is used in temperature-precipitation relationship analysis during the past decades and aided in determining likelihood") is also unclear. Just write out clearly what you mean here, even if it takes two or more sentences.

In section 2.1, I would again encourage the use of "documentary records" rather than "data."

L68: The phrase "supplementation and validation" should presumably be "to supplement or to validate." I assume the authors mean that some records were studied to provide additional information for the reconstruction (i.e., to supplement the reconstruction) while others were left out of the reconstruction in order to test the final result (i.e., to validate the reconstruction). This should be clear in the text.

L105: I think "documentary information" or "documentary evidence" would be better than "documentary proxies." Also, does ModE-RA use the same or different documentary records than those describes in section 2.1?

L106: Valler et al. 2024 does not actually use the term "time-independent." Perhaps another term should be used here, or the term should be explained. (The explanation comes in lines 201–202, but that is too late.)

L188: The phrase "were averagely deviated from climatology with around -0.5 standard deviation" sounds awkward. I bring this up as just one example of writing that the authors should correct and clarify before publication.

In Table I: The description for CWI 3 is "no record; the weather is as normal." However, the absence of a record is only a possible indicator that the weather was normal. In theory, there could be no record even for very cold or mild winters. Therefore, I would recommend a revising the description. Perhaps: "normal weather; no exceptional events for observers to record"

L204: The manuscript refers to both the CWI (in section 4.1.2) and the final posterior probability distributions of winter temperature (in section 4.2) as "the reconstruction" or "reconstructions." Moreover, it uses the acronym CWI for the cold winter indices and then uses "CWI" again in the acronyms for each of the posterior probability distributions. This can be very confusing for the reader. I recommend that the manuscript stop referring to the CWI as a "reconstruction." Moreover, the authors should find less confusing acronyms than CWI-ModE-Sim, CWI-LME, and CWI-ModE-Clim—for example, by switching "CWI" to "Post" (for posterior) (i.e., Post-ModE-Sim, etc.)

L206: "Likelihood assignment requires estimating the probability of observations being true given different indices." Although this is a correct description, I recommend that the authors explain it more clearly. Because the likelihood estimate reverses the usual approach in historical research (i.e., estimating past conditions based on present evidence) it may be counter-intuitive, and some readers will misunderstand it.

L214: From this point on, the manuscript uses the phrase "likelihood of each winter." This is a very confusing expression to a native speaker, as it would literally mean the chances of there being a winter in a given year. The authors should always spell out what they mean—i.e., the likelihood of getting the evidence for each state of winter temperature.

L229–231: This way this is written is confusing. Moreover, wouldn't it make more sense just to provide a posterior probability distribution (similar to Camenisch et al. 2022)?

L235: Rather than immediately beginning with sub-sub-section 4.1.1, I recommend that the authors first briefly explain how section 4 will be organized (under the main heading for section 4) and then how the sub-section 4.1 will be organized (under the sub-heading for section 4.1). Section 3 has already gone through the entire method up to the steps for finding the posterior probability distribution for temperature. Therefore, the manuscript must explain to the reader that section 4.1 is going back to the first step in the method (the production and analysis of the CWI). Otherwise, it looks like the manuscript is heading directly into the final temperature reconstruction. This potential confusion that is exacerbated by the manuscript's confusing use of the term "reconstruction" for the CWI, as well as the letters "CWI" in the acronyms for the final reconstructions (i.e., the posteriors).

L290–300: The authors should keep in mind that "accuracy" in conventional index methods simply means the probability that available descriptions are true representations of the weather for the season they describe. For likelihood estimates in the Bayesian method, however, we want to know both the sufficiency and necessity of each state of the weather to leave corresponding descriptions in our records. Some states of the weather (such as very cold winters) might be highly sufficient to produce corresponding descriptions but not strictly necessary (because some normal winters are described as cold). Conversely, some states of the weather (such as average winters) might be highly necessary to produce corresponding descriptions but not sufficient (because most observers didn't bother to record average conditions). Therefore, conventional terms such as "accuracy" and "reliability" may not be appropriate. It may be more appropriate to explain the value of records in terms of their power to update our knowledge about the past using a Bayesian method.

In line 315, "the *index* method" should be changed to "all index methods" since the problem applies as much or more to conventional index methods as the Bayesian method.

The image (a) in figure 4 is acceptable, but it might be redrawn to display more information and to present it more clearly. A 10-year moving average might be more informative than the decadal averages. Bars for annual CWI might be easier to read than the squiggly black line. Shading or color might possibly be used to express reliability (as described in 4.2.1).

L337: Why does the heading for subsection 4.2 say that the reconstruction begins in 1750, when it begins in 1700?

L337: Again, I would recommend that the authors include text under the sub-heading rather than beginning immediately under the sub-sub-heading 4.2.1. I recommend a couple sentences reminding readers of the meaning of "likelihoods" and their role in producing a final posterior probability distribution for temperature. Likelihoods were not explained clearly in 3.3.2, and therefore many readers will not understand the method here in section 4.2.1.

L379: For clarity, I recommend that the authors write out terms clearly. Here, for example, "posterior probability" should be "posterior probability distribution for each year's winter temperature." This approach will be new to most readers, and the manuscript should take pains to avoid vagueness or confusion.

L380–381: "Using the posterior, we reconstructed winter temperatures since 1700 and estimated their uncertainties": This statement is confusing, because the posterior probability distribution for each year's winter temperature is already a winter temperature reconstruction with uncertainties. It may be more accurate to state that the authors "used the posterior to create figure 6" or "used the posterior to visualize the best estimate and range of uncertainty for winter temperatures since 1700, as shown in figure 6."

L385–386: "In the 18th century, our reconstruction results difference about the warming in the 1760s." This sentence makes no sense. Please re-write.

Figure 6: In addition to the reconstruction of each winter's temperature, it might be interesting to compare the distribution of CWI values in figure 4(b) with the reconstructed (posterior) frequency of winters in each range. This would further illustrate how the Bayesian approach can provide a more realistic depiction of interannual temperature variability than indices alone, particularly when the documentary information is less complete or less certain.

L406: The subsection title "4.3 Source of information: observations and simulations" is vague. The subsection title and the first sentences of the subsection should explain more clearly why the authors are comparing CWI-ModE-Clim to ModE-Sim.

L417-432: While this comparison between ERA5 and ModE-Sim is interesting, it's not clear from the text what this has to do with the Bayesian reconstruction.

L446–462: When discussing the "advantages" and "limitations" of the Bayesian approach, it is important to specify what you are comparing the Bayesian approach to. The Bayesian approach has advantages and disadvantages compared to conventional historical climatology as well as advantages and disadvantages compared to paleoclimate re-analyses. Perhaps it would be helpful to present a small table comparing key features of (1) Bayesian historical climatology, (2) conventional historical climatology, and (3) paleoclimate re-analyses. Features could include, e.g., ease of use, reconstruction skill, data requirements, objectivity/subjectivity, and accurate representation of uncertainties. In any case, this section should be revised to make it clear which of the "challenges" of the Bayesian approach are already present in conventional historical climatology and which of the "advantages" are unique to the Bayesian approach.